# Predictive Markers of Treatment Response to Neoadjuvant Systemic Therapy with Dual HER2-Blockade

**DOI:** 10.3390/cancers16040842

**Published:** 2024-02-19

**Authors:** Soong June Bae, Jee Hung Kim, Min Ji Lee, Seung Ho Baek, Yoonwon Kook, Sung Gwe Ahn, Yoon Jin Cha, Joon Jeong

**Affiliations:** 1Department of Surgery, Gangnam Severance Hospital, Yonsei University College of Medicine, Seoul 06273, Republic of Korea; mission815815@yuhs.ac (S.J.B.); skymindor@yuhs.ac (M.J.L.); holydante@yuhs.ac (S.H.B.); yoonwon7@yuhs.ac (Y.K.); asg2004@yuhs.ac (S.G.A.); 2Institute for Breast Cancer Precision Medicine, Yonsei University College of Medicine, Seoul 06273, Republic of Korea; ok8504@yuhs.ac; 3Division of Medical Oncology, Department of Internal Medicine, Gangnam Severance Hospital, Yonsei University College of Medicine, Seoul 06273, Republic of Korea; 4Department of Pathology, Gangnam Severance Hospital, Yonsei University College of Medicine, Seoul 06273, Republic of Korea

**Keywords:** breast neoplasm, neoadjuvant therapy, HER2-targeted therapy, HER2 expression, tumor-infiltrating lymphocyte

## Abstract

**Simple Summary:**

In human epidermal growth factor receptor 2 (HER2)-positive breast cancer treated with neoadjuvant systemic therapy with docetaxel, carboplatin, trastuzumab, and pertuzumab (TCHP), estrogen receptor (ER) expression, HER2 protein expression, and tumor-infiltrating lymphocyte (TIL) levels are significantly associated with pathologic complete response (pCR). In ER-negative or low-ER (1–9%) breast cancer, treatment response is excellent, regardless of HER2 protein expression and TIL levels. Meanwhile, the pCR rate is notably lower in patients with ER-positive, HER2 immunohistochemistry (IHC) 2+ breast cancer, particularly in those with low TIL levels. Further investigation of novel treatment strategies for ER-positive breast cancer with HER2 IHC 2+ or low TIL levels is warranted.

**Abstract:**

In patients with human epidermal growth factor receptor 2 (HER2)-positive breast cancer, achievement of pathologic complete response (pCR) is a known prognostic indicator after neoadjuvant systemic therapy (NAST). We investigated the clinicopathological factors associated with pCR in patients with HER2-positive breast cancer treated with dual HER2-blockade. In this retrospective study, 348 patients with HER2-positive breast cancer who received NAST with docetaxel and carboplatin, combined with trastuzumab and pertuzumab (TCHP), were included. Of the 348 patients with HER2 protein expression data, 278 (79.9%) had HER2 immunochemistry (IHC) 3+. Data on tumor-infiltrating lymphocyte (TIL) levels were available for 305 patients, showing a median TIL level of 20% (IQR 5–50), among which 121 (39.7%) had high TIL levels (≥30%). Estrogen receptor (ER) status (77.9% in ER-negative vs. 47.5% in ER-positive; *p* < 0.001), HER2 protein expression (71.6% in IHC 3+ vs. 34.3% in IHC 2+; *p* < 0.001), and TIL levels (71.9% in high vs. 57.6% in low; *p* = 0.011) were significantly associated with the pCR rate. In addition, we observed a significant link between numerical TIL levels (per 10% increment) and the pCR rate. After adjusting other clinicopathologic factors, ER status (low expression [defined as 1–9% expression] or negative), HER2 IHC 3+ and numerical TIL levels (per 10% increment), and high TIL levels (≥30%) were found to be independent predictors of pCR. Notably, in ER-negative breast cancer, the treatment response was excellent, irrespective of HER2 expression and TIL levels. Conversely, in ER-positive cases, low ER expression, HER2 IHC 3+, and numerical TIL levels or high TIL levels emerged as independent predictors of pCR. Our results suggest that ER expression, HER2 protein expression, and TIL levels serve as valuable predictors of the treatment response to neoadjuvant TCHP.

## 1. Introduction

Over the past decade, the treatment paradigm for human epidermal growth factor receptor 2 (HER2)-positive breast cancer has markedly shifted. Although HER2-positive breast cancer is a biologically aggressive tumor, the revolution in HER2-targeted therapies has significantly improved the prognosis of patients with HER2-positive breast cancer [1,2].In addition, the application of neoadjuvant systemic therapy (NAST) in the early stages of HER2-positive breast cancer has increased because NAST has the benefit of reducing the surgical extent and tailoring adjuvant treatment according to its response [3,4]. As a pathologic complete response (pCR) after NAST predicts a favorable prognosis [5,6], numerous clinical trials have explored new HER2-targeted therapies to increase the pCR rate in neoadjuvant settings. Among the combinations of neoadjuvant chemotherapy and HER2-targeted therapies, the administration of a dual HER2-blockade with trastuzumab and pertuzumab to anthracycline- or platinum-based chemotherapy have shown the highest pCR rate of 50–70% [7,8,9,10,11].

Owing to its excellent response and acceptable toxicity profile, neoadjuvant chemotherapy with trastuzumab and pertuzumab is currently the preferred regimen for patients with HER2-positive breast cancer who are candidates for NAST. Although efforts have been made to identify biomarkers to predict the treatment response to neoadjuvant chemotherapy with trastuzumab and pertuzumab, it remains unclear. Trastuzumab and pertuzumab are monoclonal antibodies that bind to the extracellular subdomain of the HER2 receptor on the cancer cell surface [12]. Moreover, trastuzumab and pertuzumab cause antibody-dependent, cell-mediated cytotoxicity effects [12], indicating that immune activation in the tumor microenvironment may affect therapeutic efficacy. Based on the mechanism of action of these HER2-targeted therapies, we hypothesized that the HER2 protein expression and stromal tumor-infiltrating lymphocyte (TIL) levels evaluated by immunohistochemistry (IHC) may predict the treatment response in HER2-positive breast cancer.

This study aimed to identify the clinicopathological factors associated with pCR in patients with HER2-positive breast cancer who underwent NAST with taxane- and platinum-based chemotherapy plus trastuzumab and pertuzumab. In particular, we compared pCR rates according to estrogen receptor (ER) status, HER2 protein expression (IHC 2+ vs. IHC 3+), and TIL levels.

## 2. Patients and Methods

### 2.1. Study Population

The study protocol was reviewed and approved by the Institutional Review Board of Gangnam Severance Hospital, Yonsei University, Seoul, Korea (IRB no. 3-2023-0017). This study adhered to the principles of the Declaration of Helsinki. The requirement for written informed consent was waived because of the retrospective study design.

Between April 2015 and December 2022, we retrospectively identified 355 women with HER2-positive breast cancer who received NAST with docetaxel, carboplatin, trastuzumab, and pertuzumab (TCHP), followed by curative breast surgery, at Gangnam Severance Hospital. Patients were excluded from the analysis if they had a history of invasive breast cancer (*n* = 1), bilateral breast cancer (*n* = 4), or HER2 amplification confirmed by silver in situ hybridization (SISH) without HER2 protein expression data evaluated by IHC (*n* = 2). Finally, 348 patients were included in the study (Figure 1).

We collected clinicopathological data such as age at diagnosis, histologic grade (HG), ER status, progesterone receptor (PR) status, HER2 status, TIL levels, and clinical tumor stage and nodal status. Clinical tumor stage and nodal status were evaluated based on pretreatment multimodal imaging studies, including breast magnetic resonance imaging (MRI) and ultrasonography, which were determined according to the anatomical stage based on the 8th American Joint Committee on Cancer guidelines. All pathological data were obtained from core needle biopsy samples before NAST.

Neoadjuvant TCHP therapy was administered via intravenous infusion every 3 weeks for a total of six cycles, per the procedure in the TRYPHAENA trial [8]. Docetaxel was administered at 75 mg/m^2^ and carboplatin was administered at an area under the curve (AUC) of 5 or 6 in all cycles. The dosage of carboplatin was determined using the Calvert formula [13]. The maximum dosage was calculated using an estimated glomerular filtration rate, which was capped at 125 mL/min for patients with normal renal function. The maximum doses were calculated using the following formula: maximum carboplatin dose (mg) = target AUC (mg/mL/min) × (125 mL/min + 25). For a target AUC of 6, the maximum dose was 6 × 150 = 900 mg. For a target AUC of 5, the maximum dose was 5 × 150 = 750 mg. The physician determined the carboplatin dose based on the patient’s general characteristics, including age and underlying diseases [14]. Trastuzumab was administered at an initial dose of 8 mg/kg on day 1 of cycle 1, followed by a maintenance dose of 6 mg/kg on day 1 of cycles 2–6. Pertuzumab was administered at a loading dose of 840 mg on day 1 of cycle 1, followed by a maintenance dose of 420 mg on day 1 of cycles 2–6.

### 2.2. Immunohistochemistry and TIL Assessment

IHC staining was performed using a light microscope (BX53upright microscope; Olympus, Tokyo, Japan). Nuclear staining values of 1% or higher were considered positive for ER (clone 6F11; dilution 1:200; Leica Biosystems, Wetzlar, Germany) and PR (clone 16; dilution 1:500; Leica Biosystems, Wetzlar, Germany) [15]. ER or PR positivity ranging from 1% to 9% of invasive tumor cells was categorized as low expression. HER2 staining (clone 4B5; dilution 1:5; Ventana Medical System, Oro Valley, AZ, USA) was performed according to the 2018 American Society of Clinical Oncology/College of American Pathologists [16]. Only samples with a strong circumferential membranous HER2 immunoreactivity (3+) were considered positive, whereas samples with 0 and 1+ HER2 staining were considered negative. Patients with equivocal HER2 expression (2+) were further evaluated for HER2 gene amplification using SISH. Positive nuclear Ki-67 (clone MIB; dilution 1:1000; Abcam, Cambridge, UK) staining was assessed based on the percentage of positive tumor cells, defined as the Ki-67 labeling index.

Of the 348 patients, assessment of the TIL levels of core needle biopsy samples were available for 305 patients (87.6%, Figure 1). Stromal TIL levels were evaluated in all cores containing invasive cancer cells, according to the guidelines suggested by the International TIL Working Group [17]. Except for polymorphonuclear leukocytes, other mononuclear cells, including lymphocytes and plasma cells, were counted, and the average score was reported as a percentage [18]. Based on previous studies, a 30% cutoff was applied to divide patients into low-TIL (<30%) and high-TIL (≥30%) groups [19,20,21].

### 2.3. Statistical Analyses

The objective of this study was to assess the predictive value of ER expression, HER2 protein expression, and TIL levels evaluated using IHC for treatment response in patients with HER2-positive breast cancer who received neoadjuvant TCHP therapy. We evaluated the pCR rate according to ER status, HER2 protein expression, and TIL levels. pCR was defined as no evidence of invasive breast cancer residues in either the breast or the axillary lymph nodes (ypT0/is, ypN0). Continuous variables were compared using Student’s *t*-test, and categorical variables were compared using the chi-squared test or Fisher’s exact test. Univariable and multivariable analyses were performed using a binary logistic regression model to identify the predictive clinicopathological features of pCR. Factors considered in the multivariable analysis included age (<50 vs. ≥50), HG (1 or 2 vs. 3), ER expression (0% vs. 1–9% vs. ≥10%), PR expression (0% vs. 1–9% vs. ≥10%), TIL levels as continuous values (per 10% increment) or categorical values (<30% vs. ≥30%), clinical tumor stage (1 vs. 2 vs. ≥3), and clinical nodal status (negative vs. positive). Ki-67 was not included in the multivariable analysis because this value was evaluated in a small number of patients. Odds ratios (ORs) and 95% confidence intervals (CIs) with two-sided *p*-values were calculated. All tests were two-sided, and *p*-values less than 0.05 were considered statistically significant. All statistical analyses were performed using SPSS version 25 software (SPSS, Armonk, NY, USA).

## 3. Results

### 3.1. Baseline Characteristics

A total of 348 women with HER2-positive breast cancer who underwent neoadjuvant TCHP followed by curative surgery were identified. The baseline characteristics of the patients are described in Table 1. Age ≥ 50 years was observed in 179 (51.4%) patients, and 158 (45.4%) patients were hormone receptor (HR)-positive. All HR-positive patients were ER-positive, but 41.1% (65 of 158) were PR-negative. The majority of patients had HG 1-2 (249 of 304 [86.8%]) and clinical T2 or T3 stage (339 of 348 [97.4%]), and were clinically node-positive (279 of 348 [80.2%]). Ki-67 in pre-treatment biopsy samples was evaluated in 60 (17.2%) of the 348 patients, and the median value was 30% (IQR 20–60). In addition, TIL levels in pre-treatment biopsy samples were evaluated in 305 (87.6%) of the 348 patients, and the median value was 20% (IQR 5–50).

Of the 348 patients, 278 (79.9%) had HER2 IHC 3+ and 70 (20.1%) had HER2 IHC 2+, with HER2 amplification detected using SISH. Patients with HER2 IHC 3+ had a significantly higher proportion of ER-negative (62.6% vs. 22.9%, *p* < 0.001) and PR-negative (79.1% vs. 50.0%, *p* < 0.001) tumors than those with HER2 IHC 2+. Compared to patients with HER2 IHC 2+, clinical stage T3 disease or higher was more frequently observed in patients with HER2 IHC 3+, although the difference was not statistically significant (44.6% vs. 30.0%, *p* = 0.085). The other characteristics, including age, HG, TIL levels, clinical nodal status, and Ki-67, did not differ according to HER2 IHC expression.

Among the patients with available TIL levels, 121 (39.7%) had high TIL levels and 184 (60.3%) had low TIL levels (Appendix A). Patients with high TIL levels were more likely to have high HG (27.1% vs. 12.5%, *p* = 0.001) and an ER-negative status (61.2% vs. 46.2%, *p* = 0.011) than those with low TIL levels. Meanwhile, the proportion of patients with clinical stage T3 disease or higher was lower in the high-TIL-level group than in the low-TIL-level group (29.8% vs. 45.1%, *p* = 0.026).

### 3.2. Pathologic Complete Response Related Factors

The overall pCR rate was 65.6% (228 of 348 patients). ER status (ER-negative, 77.9% [148 of 190] vs. ER-positive, 47.5% [75 of 158]; *p* < 0.001), HER2 IHC expression (IHC 3+, 71.6% [199 of 278] vs. IHC 2+, 34.3% [24 of 70]; *p* < 0.001), and TIL levels (high TIL levels, 71.9% [87 of 121] vs. low TIL levels, 57.6% [106 of 184]; *p* = 0.011) were significantly associated with pCR (Figure 2). In addition, we found a significant correlation between numerical TIL levels (per 10% increment) and pCR (OR, 1.12; 95% CI, 1.03–1.22; *p* = 0.008; Table 2). Moreover, the multivariable analysis including numerical TIL levels (per 10% increment) as a confounding factor (multivariable model 1 in Table 2) revealed that low ER expression or ER negativity (1–9%, OR, 3.47; 95% CI, 1.59–7.61; *p* = 0.002; 0%, OR, 5.23; 95% CI, 1.53–17.86; *p* = 0.008), HER2 IHC 3+ (OR, 3.07; 95% CI, 1.57–6.00; *p* = 0.001), and numerical TIL levels (OR, 1.11; 95% CI, 1.01–1.23; *p* = 0.040) were independent predictive factors for pCR. The multivariable model discriminated the data well, with an AUC of 0.782. With an increase in TIL levels of 10%, the odds for pCR increased by 11%. Similarly, in the multivariable analysis including high or low TIL levels (cut-off value of 30%) as a confounding factor (multivariable model 2 in Table 2), low ER expression or ER negativity (1–9%, OR, 3.42; 95% CI, 1.56–7.48; *p* = 0.002; 0%, OR, 5.25; 95% CI, 1.54–17.93; *p* = 0.008), HER2 IHC 3+ (OR, 3.09; 95% CI, 1.58–6.05; *p* = 0.001), and high TIL levels (OR, 1.88; 95% CI, 1.04–3.40; *p* = 0.037) were independent predictive factors for pCR. The multivariable model performed well in terms of discriminating the data, achieving an AUC of 0.780.

Next, we assessed whether the predictive values for pCR differed according to the ER status. In patients with ER-negative breast cancer, there was no difference in the pCR rate according to HER2 IHC expression (78.7% [IHC 3+] vs. 68.8% [IHC 2+], *p* = 0.354; Figure 3A) or TIL levels (79.7% [high TIL levels] vs. 78.8% [low TIL levels], *p* = 0.888; Figure 3B). In contrast, among the patients with ER-positive breast cancer, the pCR rate was significantly higher in the groups with HER2 IHC 3+ (59.6% vs. 24.1%; *p* < 0.001; Figure 4A) and high TIL levels (59.6% vs. 39.4%, *p* = 0.022; Figure 4B) than that in the groups with HER2 IHC 2+ and low TIL levels, respectively. Furthermore, pCR was more frequently observed in the low-ER-expression group than in the high-ER-expression group (84.6% vs. 40.2%, *p* = 0.022; Figure 4C). Upon multivariable analysis including numerical TIL levels (per 10% increment) as a confounding factor (multivariable model 1 in Table 3), HER2 IHC 3+ (OR, 4.69; 95% CI, 1.98–11.09; *p* < 0.001), numerical TIL levels (OR, 1.22; 95% CI, 1.04–1.42; *p* = 0.013), and low ER expression (1–9%, OR, 4.92; 95% CI, 1.36–17.84; *p* = 0.015) remained significant predictors for pCR in ER-positive breast cancer. The multivariable model discriminated the data well, with an AUC of 0.784. With a 10% increase in TIL levels, the odds of achieving pCR increased by 22%. Likewise, the multivariable analysis including high or low TIL levels (cut-off value of 30%) as a confounding factor (multivariable model 2 in Table 3) showed that HER2 IHC 3+ (OR, 4.93; 95% CI, 2.06–11.82; *p* < 0.001), high TIL levels (OR, 3.24; 95% CI, 1.37–7.66; *p* = 0.007), and low ER expression (1–9%, OR, 4.96; 95% CI, 1.36–18.07; *p* = 0.015) were independent predictive factors for pCR in ER-positive breast cancer. The multivariable model performed well in terms of discriminating the data, achieving an AUC of 0.780.

We evaluated the pCR rate according to HER2 expression and ER expression, stratified by TIL levels, in ER-positive, HER2-positive breast cancer (Figure 5). Notably, the pCR rate was about 80% in the low-ER-expression group, regardless of TIL levels. High TIL levels were significantly associated with pCR in the HER2 IHC 3+ group (77.8% vs. 50.8%, *p* = 0.017) and the high-ER-expression group (53.8% vs. 32.9%, *p* = 0.027). Additionally, among the ER-positive, HER2 IHC 2+ group, the pCR rate tended to be higher in patients with high TIL levels than in those with low TIL levels, but the difference was not significant (35% vs. 17.6%, *p* = 0.150).

## 4. Discussion

In this study, we found that ER expression, HER2 IHC 3+ status, and high TIL levels were independent predictors of high pCR rates in patients with HER2-positive breast cancer who underwent neoadjuvant TCHP. Accumulating evidence suggests that the HER2-enriched intrinsic subtype and immune system features are the most validated predictive biomarkers for HER2-targeted therapy. Previous clinical trials have shown that the treatment response to HER2-targeted therapy is higher in the HER2-enriched subtype [22,23,24,25,26,27]. Similarly, Memorial Sloan Kettering Cancer Center data have demonstrated that higher HER2 protein expression levels show a significant intent to achieve pCR (66% in the HER2 IHC 3+ group vs. 17% in the HER2 IHC 2+ group with confirmed HER2 amplification) [28], probably derived from a higher proportion of the HER2-enriched subtype within the HER2 IHC 3+ group. With respect to immune system features, the presence of TILs, programmed death ligand-1 protein expression, T-cell receptor diversity metrics, and immune-related gene signatures are associated with the NAST response [26,27,29,30,31].

As is consistent with the previous literature [5,7,8], another significant factor related to pCR was ER status; the pCR rate was 77.9% in ER-negative breast cancer and 47.5% in ER-positive breast cancer. However, data on the effects of HER2 protein expression and TIL levels on pCR rates according to the ER status are limited. Our study revealed that the predictive value of HER2 protein expression and stromal TIL levels differed according to ER status. HER2 IHC 3+ and high TIL levels were significantly associated with high pCR rates in ER-positive breast cancer. Meanwhile, patients with ER-negative breast cancer achieved an excellent treatment response, regardless of HER2 protein expression and TIL levels. Interestingly, a high pCR rate was also evident in patients with low ER expression (1–9%), irrespective of HER2 protein expression and TIL levels, aligning with the classification of low-ER-expression, HER2- negative breast cancer as a triple-negative breast cancer [32,33,34,35].

A possible explanation for these findings could be the proportional difference in the HER2-enriched subtype according to the ER status and HER2 protein levels [36]. The HER2-enriched subtype is found in 80–90% of ER-negative and HER2-positive breast cancers and 20–50% of ER-positive and HER2-positive breast cancer [25,37]. In this study, the majority of patients with ER-negative breast cancer (174 of 190, 91.6%) were HER2 IHC 3+, estimated to be of the HER2-enriched subtype, which may lead to an excellent response to TCHP regardless of TIL levels, considering the mechanisms of trastuzumab and pertuzumab [12]. The lack of difference in the pCR rate between HER2 IHC 2+ and 3+ tumors may imply a strong oncogenic addiction of HER2-positive breast cancer and the efficacy of dual HER2 blockade in ER-negative breast cancer.

Historically, baseline TIL levels have been considered a predictor of NAST in HER2-positive breast cancer [38,39]. However, the predictive role of TIL levels remains inconclusive when limited to patients receiving NAST-containing trastuzumab and pertuzumab. In exploratory analyses from NeoSphere and TRYPHAENA, immune-related gene signatures were predictive of pCR, but the TIL levels itself were not [31,40]. In the present study, we found that TIL levels were specifically predictive of pCR in ER-positive breast cancer patients treated with neoadjuvant TCHP. Of these, patients with ER-positive breast cancer with HER2 IHC 3+ and high TIL levels exhibited a response rate similar to that of patients with ER-negative, HER2-positive breast cancer. A previous investigation utilizing the PAM50 classification to define the intrinsic subtypes of HER2-positive breast cancer revealed the highest TIL levels in the HER2-enriched subtype [26]. Thus, high HER2 expression and TIL levels may serve as indicators of the HER2-enriched subtype of ER-positive breast cancer.

Notably, the pCR rate was significantly lower in patients with ER-positive, HER2 IHC 2+ breast cancer, particularly in those with low TIL levels. These patients could be a specific population requiring a new treatment strategy to improve the treatment response. Recently, a novel antibody–drug conjugate, trastuzumab deruxtecan, was approved for the treatment of metastatic HER2-positive breast cancer [41]. Furthermore, trastuzumab deruxtecan is effective in patients with HER2-low (IHC 1+ or 2+ without amplification) breast cancer via the bystander effect [42,43]. The DESTINY-Breast04 trial showed that trastuzumab deruxtecan had superior therapeutic efficacy compared to standard chemotherapy for HER2-low metastatic breast cancer [44]. Considering that most patients (approximately 89%) enrolled in the DESTINY-Breast04 trial were ER-positive, this new targeted agent could be a promising alternative in patients with ER-positive, HER2 IHC 2+ breast cancer.

Our study has several limitations. First, as this was a retrospective study conducted at a single institution, our results need to be validated in an independent cohort. Second, the number of subpopulations according to ER, HER2 expression, and TIL levels was relatively small. Specifically, only 13 patients had ER-negative, HER2 IHC 2+ breast cancer. Third, Ki-67 was assessed in only 60 patients (17.2%) of the entire cohort because this parameter was not routinely evaluated via core needle biopsy samples at our institution. Consequently, we excluded Ki-67 from multivariable analysis to maintain statistical power, despite its recognized role as a predictive factor for pCR [39,45]. Fourth, we did not perform a genomic analysis that could have elucidated the differences in genomic signatures based on ER status or TIL levels. Finally, the short median follow-up period of our study cohort precluded the investigation of prognostic factors. Despite these limitations, to the best of our knowledge, our study has the strength of exploring predictive clinicopathological factors with the largest number of patients who received homogeneous treatment with taxane and platinum-based standard chemotherapy plus trastuzumab and pertuzumab.

In summary, our findings indicate that TCHP shows an excellent response in patients with ER-negative, HER2-positive breast cancer, regardless of HER2 protein expression and TIL levels. Meanwhile, HER2 protein expression and TIL levels are independent predictors of treatment response in ER-positive breast cancer. Remarkably, among the patients with ER-positive, HER2-positive breast cancer, those with low ER expression or high HER2 expression alongside elevated TIL levels exhibited response rates comparable to patients with ER-negative, HER2-positive breast cancer. However, the remaining subgroup within the ER-positive population derived few benefits from neoadjuvant TCHP, underscoring the need for novel treatment strategies tailored to these specific subpopulations.

## Figures and Tables

**Figure 1 cancers-16-00842-f001:**
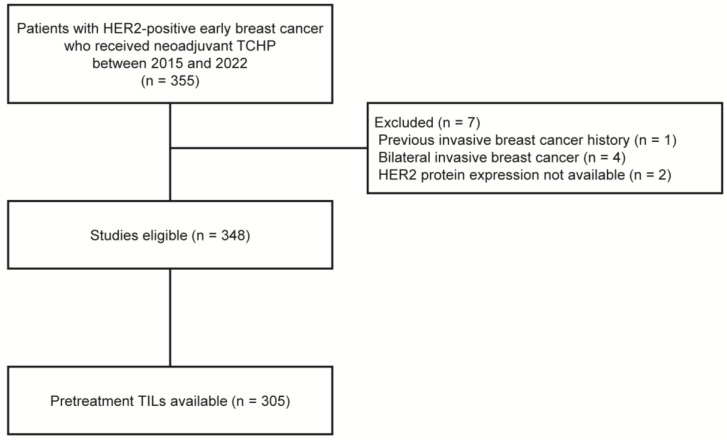
Flow diagram of patients. HER2, human epidermal growth factor receptor 2; TCHP, docetaxel, carboplatin, trastuzumab, and pertuzumab; TILs, tumor-infiltrating lymphocytes.

**Figure 2 cancers-16-00842-f002:**
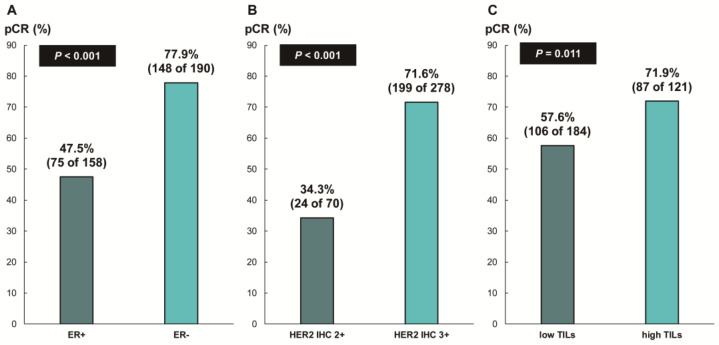
Pathologic complete response according to (**A**) ER status, (**B**) HER2 immunohistochemistry (IHC) expression, and (**C**) tumor-infiltrating lymphocytes (TILs). ER, estrogen receptor; HER2, human epidermal growth factor receptor 2.

**Figure 3 cancers-16-00842-f003:**
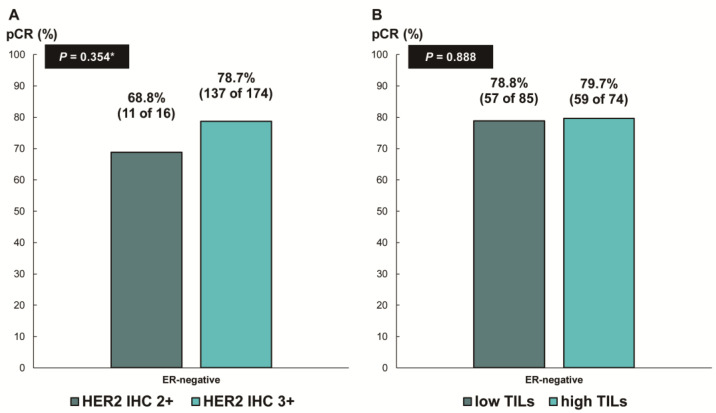
Pathologic complete response according to (**A**) HER2 immunohistochemistry (IHC) expression and (**B**) tumor-infiltrating lymphocytes (TILs) in ER-negative, HER2-positive breast cancer. HER2, human epidermal growth factor receptor 2; ER, estrogen receptor. * *p*-value was obtained using Fisher’s exact test.

**Figure 4 cancers-16-00842-f004:**
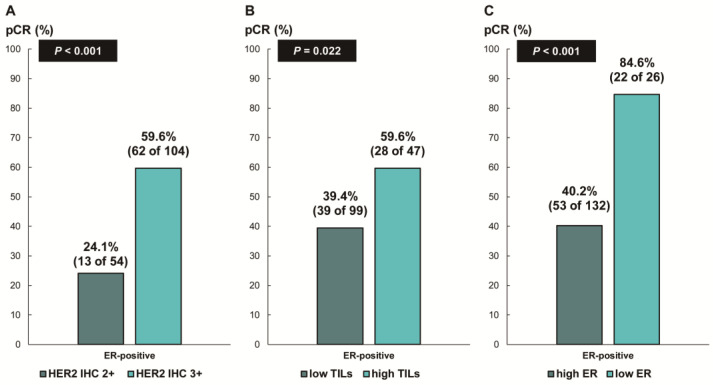
Pathologic complete response according to (**A**) HER2 immunohistochemistry (IHC) expression; (**B**) tumor-infiltrating lymphocytes (TILs); and (**C**) ER expression in ER-positive, HER2-positive breast cancer. HER2, human epidermal growth factor receptor 2; ER, estrogen receptor.

**Figure 5 cancers-16-00842-f005:**
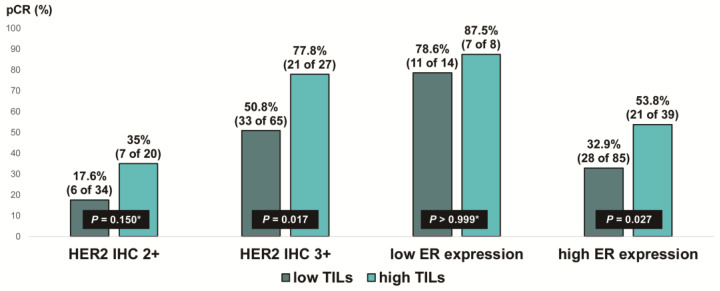
Pathologic complete response according to HER2 immunohistochemistry (IHC) expression and ER expression in ER-positive, HER2-positive breast cancer stratified by tumor-infiltrating lymphocyte (TIL) levels. HER2, human epidermal growth factor receptor 2; ER, estrogen receptor. * *p*-value was obtained using Fisher’s exact test.

**Table 1 cancers-16-00842-t001:** Baseline characteristics of patients according to HER2 expression.

Variables	HER2 IHC 2+	HER2 IHC 3+	Total	*p* Value
Age				0.790
<50	33 (47.1)	136 (48.9)	169 (48.6)	
≥50	37 (52.9)	142 (51.1)	179 (51.4)	
HG *				0.238
1 or 2	59 (86.8)	190 (80.5)	249 (81.9)	
3	9 (13.2)	46 (19.5)	55 (18.1)	
HR				<0.001
positive	54 (77.1)	104 (37.4)	158 (45.4)	
negative	16 (22.9)	174 (62.6)	190 (54.6)	
ER expression				<0.001
≥10%	51 (72.9)	81 (29.1)	132 (37.9)	
1–9%	3 (4.3)	23 (8.3)	26 (7.5)	
0	16 (22.9)	174 (62.6)	190 (54.6)	
PR expression				<0.001
≥10%	30 (42.9)	36 (12.9)	66 (19.0)	
1–9%	5 (7.1)	22 (7.9)	27 (7.8)	
0	35 (50.0)	220 (79.1)	255 (73.3)	
TIL levels * (%), median (IQR)	10 (5–60)	20 (8.75–50)	20 (5–50)	0.729
TIL levels *				0.870
<30%	41 (61.2)	143 (60.1)	184 (60.3)	
≥30%	26 (38.8)	95 (39.9)	121 (39.7)	
Clinical tumor stage				0.085
1	2 (2.9)	7 (2.5)	9 (2.6)	
2	47 (67.1)	147 (52.9)	194 (55.7)	
≥3	21 (30.0)	124 (44.6)	145 (41.7)	
Clinical nodal status				0.768
negative	13 (18.6)	56 (20.1)	69 (19.8)	
positive	57 (81.4)	222 (79.9)	279 (80.2)	
Ki-67 * (%), median (IQR)	35 (20–65)	30 (20–60)	30 (20–60)	0.743

Note: Unless otherwise noted, values represent the number of patients, with percentages in parentheses. * Missing value. HER2 = human epidermal growth receptor factor 2, IHC = immunohistochemistry, HG = histologic grade, HR = hormone receptor, ER = estrogen receptor, PR = progesterone receptor, TIL = tumor-infiltrating lymphocyte.

**Table 2 cancers-16-00842-t002:** Odds ratio (OR) and 95% confidence interval (CI) for pCR in all patients (multivariable model 1, AUC = 0.782, log likelihood ratio test *p* < 0.001; multivariable model 2, AUC = 0.780, log likelihood ratio test *p* < 0.001).

Variables	Univariable		Multivariable Model 1		Multivariable Model 2	
	OR (95% CI)	*p* Value	OR (95% CI)	*p* Value	OR (95% CI)	*p* Value
Age						
<50	1 (Reference)		1 (Reference)		1 (Reference)	
≥50	1.68 (1.08–2.61)	0.022	1.21 (0.69–2.13)	0.515	1.18 (0.67–2.08)	0.560
HG						
1 or 2	1 (Reference)		1 (Reference)		1 (Reference)	
3	1.33 (0.71–2.49)	0.369	0.73 (0.35–1.53)	0.402	0.76 (0.36–1.58)	0.457
ER expression		<0.001		0.002		0.002
≥10%	1 (Reference)		1 (Reference)		1 (Reference)	
1–9%	5.25 (3.22–8.56)	<0.001	3.47 (1.59–7.61)	0.002	3.42 (1.56–7.48)	0.002
0	8.20 (2.67–25.15)	<0.001	5.23 (1.53–17.86)	0.008	5.25 (1.54–17.93)	0.008
PR expression		<0.001		0.500		0.483
≥10%	1 (Reference)		1 (Reference)		1 (Reference)	
1–9%	4.81 (2.71–8.54)	<0.001	1.67 (0.71–3.91)	0.240	1.68 (0.72–3.94)	0.231
0	1.40 (0.56–3.48)	0.469	1.22 (0.45–3.33)	0.697	1.19 (0.44–3.26)	0.735
HER2						
IHC 2+	1 (Reference)		1 (Reference)		1 (Reference)	
IHC 3+	4.83 (2.76–8.44)	<0.001	3.07 (1.57–6.00)	0.001	3.09 (1.58–6.05)	0.001
TIL levels (per 10% increment)	1.12 (1.03–1.22)	0.008	1.11 (1.01–1.23)	0.040	-	-
TIL levels						
<30%	1 (Reference)		-	-	1 (Reference)	
≥30%	1.88 (1.15–3.08)	0.012	-	-	1.88 (1.04–3.40)	0.037
Clinical tumor stage		0.599		0.405		0.401
1	1 (Reference)		1 (Reference)		1 (Reference)	
2	0.53 (0.11–2.62)	0.436	0.55 (0.10–3.23)	0.504	0.51 (0.09–3.08)	0.464
≥3	0.47 (0.09–2.33)	0.354	0.39 (0.07–2.40)	0.312	0.37 (0.06–2.31)	0.289
Clinical nodal status						
negative	1 (Reference)		1 (Reference)		1 (Reference)	
positive	0.62 (0.35–1.11)	0.107	0.58 (0.28–1.19)	0.135	0.58 (0.28–1.18)	0.133
Ki-67 * (%)	1.01 (0.99–1.03)	0.526	-	-	-	-

* Ki-67 was not included in the multivariable model because it was evaluated in a small number of patients. pCR = pathologic complete response, HG = histologic grade, ER = estrogen receptor, PR = progesterone receptor, HER2 = human epidermal growth receptor factor 2, IHC = immunohistochemistry, TIL = tumor-infiltrating lymphocyte.

**Table 3 cancers-16-00842-t003:** Odds ratio (OR) and 95% confidence interval (CI) for pCR in patients with ER + HER2+ breast cancer (multivariable model 1, AUC = 0.784, log likelihood ratio test *p* < 0.001; multivariable model 2, AUC = 0.782, log likelihood ratio test *p* < 0.001).

Variables	Univariable		Multivariable Model 1		Multivariable Model 2	
	OR (95% CI)	*p* Value	OR (95% CI)	*p* Value	OR (95% CI)	*p* Value
Age						
<50	1 (Reference)		1 (Reference)		1 (Reference)	
≥50	1.32 (0.70–2.49)	0.389	1.47 (0.65–3.31)	0.353	1.45 (0.64–3.27)	0.371
HG						
1 or 2	1 (Reference)		1 (Reference)		1 (Reference)	
3	0.86 (0.28–2.61)	0.789	0.51 (0.13–1.99)	0.512	0.61 (0.16–2.28)	0.459
ER expression						
≥10%	1 (Reference)		1 (Reference)		1 (Reference)	
1–9%	8.20 (2.67–25.15)	<0.001	4.92 (1.36–17.84)	0.015	4.96 (1.36–18.07)	0.015
PR expression		0.026		0.738		0.718
≥10%	1 (Reference)		1 (Reference)		1 (Reference)	
1–9%	2.63 (1.30–5.32)	0.007	1.42 (0.85–3.47)	0.438	1.44 (0.59–3.51)	0.426
0	1.40 (0.56–3.48)	0.469	1.13 (0.40–3.18)	0.824	1.07 (0.38–3.07)	0.895
HER2						
IHC 2+	1 (Reference)		1 (Reference)		1 (Reference)	
IHC 3+	4.66 (2.23–9.73)	<0.001	4.69 (1.98–11.09)	<0.001	4.93 (2.06–11.82)	<0.001
TIL levels (per 10% increment)	1.14 (1.01–1.30)	0.034	1.22 (1.04–1.42)	0.013	-	-
TIL levels						
<30%	1 (Reference)		-	-	1 (Reference)	
≥30%	2.27 (1.12–4.60)	0.024	-	-	3.24 (1.37–7.66)	0.007
Clinical tumor stage		0.640		0.980		0.953
1	1 (Reference)		1 (Reference)		1 (Reference)	
2	0.54 (0.09–3.36)	0.505	0.89 (0.11–7.15)	0.850	0.75 (0.09–5.97)	0.783
≥3	0.69 (0.11–4.42)	0.694	0.83 (0.10–7.19)	0.867	0.71 (0.08–6.09)	0.758
Clinical nodal status						
negative	1 (Reference)		1 (Reference)		1 (Reference)	
positive	0.96 (0.43–2.15)	0.923	0.75 (0.28–2.01)	0.572	0.76 (0.28–2.02)	0.576
Ki-67 * (%)	1.00 (0.96–1.04)	0.919	-	-	-	-

* Ki-67 was not included in the multivariable model because it was evaluated in a small number of patients. pCR = pathologic complete response, HG = histologic grade, HER2 = human epidermal growth receptor factor 2, ER = estrogen receptor, PR = progesterone receptor, IHC = immunohistochemistry, TIL = tumor-infiltrating lymphocyte.

## Data Availability

The datasets generated and analyzed during the current study are available from the corresponding author on request.

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
