# Peer review of "Predictive Markers of Treatment Response to Neoadjuvant Systemic Therapy with Dual HER2-Blockade"

_cancers, 2024, doi:10.3390/cancers16040842_

Round 1
Reviewer 1 Report
Comments and Suggestions for Authors
I am glad to have the opportunity to review the article entitled 'Predictive markers of treatment response to neoadjuvant systemic therapy with dual HER2-blockade'.
In the article authors represent a retrospective analysis of predictive factors for pCR in HER2-positive early breast cancer.
They have relatively large sample of patients (n=348, of them 305 for multivariate analysis), all were treated with the same chemotherapy (docetaxel and carboplatin) and dual HER2 blockade. In addition to cT, cN, grade and age they studied the expression of HER2 (IHC2+ vs IHC 3+) and tumor lymphocyte infiltration (TIL) on pCR.
There is already known that HER2+ subtype has high pCR rate and that HER2+ HR+ subtype has lower percentage of pCR than HER2+ HR-.
The only new data could be about TIL and Ki-67.
My comments to improve the article:
1. Ki-67 method was introduced in the methods, however there are no data on that in the article. Indeed, it would be of importance to include Ki-67 into the model for pCR. I suggest to present in results also data about the distribution the Ki -67 (median.; Q1, Q3 etc.).
2. TIL is presented as high or low. As for Ki-67, something should be said about distribution (normal or not) and to define the measures of distribution. It would be useful to say more clearly why the cut-off of 30% was chosen. It exists no international recommendation about this. My suggestion, however, is to present TIL as a continuous variable and consider how much an increase in the percentage of TIL increases the pCR (e.g. a 10% increase in TIL increases the pCR by 20%).
3. Instead to evaluate prognostic value of TIL according to ER status and HER2 by Fisher Exact test, ER should be added into multivariate model as a separate variable. This would show, if role of ER is prognostic when correcting for all other factors.
4. For a multivariable logistic model, please add AUC value and log likelihood ratio test.
5. Figure 4: It should be indicated to which comparison refers each p-value; It would be more understandable, it the bar for a total TIL before each group would be removed (in case the above suggested change will be done, Figure 4 may be removed from the article and replaced by another).
6. Some recent prospective studies on TIL (published in Cancers and other per reviewed journals) could be included in discussion.
Comments on the Quality of English LanguageThe quality of English language is good.
Author Response
<Reviewer 1>
I am glad to have the opportunity to review the article entitled 'Predictive markers of treatment response to neoadjuvant systemic therapy with dual HER2-blockade'.
In the article authors represent a retrospective analysis of predictive factors for pCR in HER2-positive early breast cancer.
They have relatively large sample of patients (n=348, of them 305 for multivariate analysis), all were treated with the same chemotherapy (docetaxel and carboplatin) and dual HER2 blockade. In addition to cT, cN, grade and age they studied the expression of HER2 (IHC2+ vs IHC 3+) and tumor lymphocyte infiltration (TIL) on pCR.
There is already known that HER2+ subtype has high pCR rate and that HER2+ HR+ subtype has lower percentage of pCR than HER2+ HR-.
The only new data could be about TIL and Ki-67.
We are sincerely grateful for the time and thoughtful consideration you have dedicated to reviewing this manuscript. We acknowledge the constructive feedback provided by the reviewers, and we have incorporated many of their suggestions to improve clarity. Your valuable input has significantly contributed to enhancing the quality and precision of our manuscript. Thank you once again for your invaluable contributions.
My comments to improve the article:
- Ki-67 method was introduced in the methods, however there are no data on that in the article. Indeed, it would be of importance to include Ki-67 into the model for pCR. I suggest to present in results also data about the distribution the Ki -67 (median.; Q1, Q3 etc.).
Answer) We appreciate your valuable comment. According to your suggestion, we added Ki-67 values (median, IQR) in the revised manuscript (Results, Table 1, and Supplementary Table 1). While Ki-67 is indeed a recognized predictive factor for pCR, it is not routinely assessed in core needle biopsy samples at our institution. As a result, Ki-67 was evaluated in approximately 17% of the entire cohort, leading to its exclusion from multivariable analysis to ensure statistical power. Also, we have included a discussion on this limitation in the “Discussion” section:
Third, Ki-67 was assessed in only 60 patients (17.2%) of the entire cohort because this parameter was not routinely evaluated in core needle biopsy samples at our institution. Consequently, we excluded Ki-67 from multivariable analysis to maintain statistical power, despite its recognized role as a predictive factor for pCR.
- TIL is presented as high or low. As for Ki-67, something should be said about distribution (normal or not) and to define the measures of distribution. It would be useful to say more clearly why the cut-off of 30% was chosen. It exists no international recommendation about this. My suggestion, however, is to present TIL as a continuous variable and consider how much an increase in the percentage of TIL increases the pCR (e.g. a 10% increase in TIL increases the pCR by 20%).
- Instead to evaluate prognostic value of TIL according to ER status and HER2 by Fisher Exact test, ER should be added into multivariate model as a separate variable. This would show, if role of ER is prognostic when correcting for all other factors.
Answer for 2-3) Thank you for your insightful feedback. As with Ki-67, we presented the median and IQR values of TIL levels (revised Table 1 and Results section). We determined the cut-off value for TILs at 30%, considering the results of prior studies and the distribution of patients assigned to the high TILs and low TILs groups. We have also cited the related references in the revised manuscript. In addition, we have confirmed that both the continuous value (per 10% increment, multivariable model 1 in revised Table 2 and 3) and categorical value (≥ 30% vs. < 30%, multivariable model 2 in revised Table 2 and 3) of TIL levels were independent factors related to pCR, respectively.
Originally, hormone-receptor (HR) was included in the multivariable model. In this study, all patients with HR-positive breast cancer were estrogen receptor (ER)-positive. To reduce confusion, we replaced HR to ER in the entire revised manuscript. Following Reviewer 2’s suggestion, we included ER expression (≥ 10% vs. 1-9% vs. 0) and progesterone receptor (PR) expression (≥ 10% vs. 1-9% vs. 0) in the multivariable model, instead of HR. Ultimately, we found that low ER expression or ER-negativity, HER2 IHC expression, and TIL levels (both continuous or categorical values) were independent factors for pCR in all patients. Similarly, low ER expression, HER2 IHC expression, and TIL levels (both continuous or categorical values) emerged as independent factors for pCR in patients with ER-positive, HER2-positive breast cancer.
We have ensured that all relevant sections throughout the manuscript reflect these revisions accordingly.
- For a multivariable logistic model, please add AUC value and log likelihood ratio test.
Answer) Thank you for your comment. We added AUC value and log likelihood ratio test of each multivariable logistic model.
- Figure 4: It should be indicated to which comparison refers each p-value; It would be more understandable, it the bar for a total TIL before each group would be removed (in case the above suggested change will be done, Figure 4 may be removed from the article and replaced by another).
Answer) Considering other reviewers’ suggestions, we modified figure 2-4 as follows:
Figure 2. Pathologic complete response according to (A) ER status, (B) HER2 immunohistochemistry (IHC) expression and (C) tumor-infiltrating lymphocytes (TILs). ER, estrogen receptor; HER2, human epidermal growth factor receptor 2
Figure 3. Pathologic complete response according to (A) HER2 immunohistochemistry (IHC) expression and (B) tumor-infiltrating lymphocytes (TILs) in ER-negative, HER2-positive breast cancer. HER2, human epidermal growth factor receptor 2; ER, estrogen-receptor. *p-value was obtained with the Fisher’s exact test.
Figure 4. Pathologic complete response according to (A) HER2 immunohistochemistry (IHC) expression, (B) tumor-infiltrating lymphocytes (TILs) and (C) ER expression in ER-positive, HER2-positive breast cancer. HER2, human epidermal growth factor receptor 2; ER, estrogen-receptor.
Figure 5. Pathologic complete response according to HER2 immunohistochemistry (IHC) expression and ER expression in ER-positive, HER2-positive breast cancer stratified by tumor-infiltrating lymphocytes (TILs). HER2, human epidermal growth factor receptor 2; ER, estrogen receptor. *p-value was obtained with the Fisher’s exact test.
- Some recent prospective studies on TIL(published in Cancers and other per reviewed journals) could be included in discussion.
Answer) Following your suggestion, we have cited new references (38, 39) in the “Discussion” section.
- Wein, L.; Savas, P.; Luen, S.J.; Virassamy, B.; Salgado, R.; Loi, S. Clinical validity and utility of tumor-infiltrating lymphocytes in routine clinical practice for breast cancer patients: Current and future directions. Front Oncol 2017, 7, 156.
- Geršak, K.; Geršak, B.M.; Gazić, B.; Klevišar Ivančič, A.; Drev, P.; Ružić Gorenjec, N.; Grašič Kuhar, C. The possible role of anti- and protumor-infiltrating lymphocytes in pathologic complete response in early breast cancer patients treated with neoadjuvant systemic therapy. Cancers (Basel) 2023, 15.

Reviewer 2 Report
Comments and Suggestions for Authors
Overall, this is an excellent retrospective analysis. Just a few comments/questions:
1) Would it be possible to analyze the impact of level of ER expression on pCR rates, not just ER+ (>1% vs. <1%) vs. ER-? As we're realizing that patients with ER 1-9%/HER2- behave like (and should be treated as) TNBC, it would be interesting to know if ER 1-9%/HER2+ respond like ER-/HER2+ (or not). Even if this subgroup is too small to analyze it would be nice to know this.
2) In your methods section you mention assessing Ki-67 labelling index on the patients' cancers, but you never mention it again. Is this because it did not impact pCR rates or the impact of HR status, HER2 expression, or TIL level on pCR rates? If so, this would be worth mentioning.
3) There is a major typo on page 6. line 190 - you're reporting the impact of high or low TILs on pCR rates in HR-negative patients but you mistakenly repeat IHC3+ vs. IHC2+ instead.
4) Please increase the font on the legend for Figure 4 - this is your most important data - don't want readers to have trouble seeing it.
Comments on the Quality of English LanguageNone
Author Response
<Reviewer 2>
Overall, this is an excellent retrospective analysis. Just a few comments/questions:
We are sincerely grateful for the time and thoughtful consideration you have dedicated to reviewing this manuscript. We acknowledge the constructive feedback provided by the reviewers, and we have incorporated many of their suggestions to improve clarity. Your valuable input has significantly contributed to enhancing the quality and precision of our manuscript. Thank you once again for your invaluable contributions.
1) Would it be possible to analyze the impact of level of ER expression on pCR rates, not just ER+ (>1% vs. <1%) vs. ER-? As we're realizing that patients with ER 1-9%/HER2- behave like (and should be treated as) TNBC, it would be interesting to know if ER 1-9%/HER2+ respond like ER-/HER2+ (or not). Even if this subgroup is too small to analyze it would be nice to know this.
Answer) We appreciate your thorough review and the valuable comment. First of all, we have replaced “hormone-receptor (HR)” with “estrogen-receptor (ER)” throughout the revised manuscript to enhance clarity, considering that all patients with HR-positive breast cancer were ER-positive.
According to your suggestion, we have re-evaluated the impact of ER expression (≥ 10% vs. 1-9% vs. 0) on pCR, particularly in ER-positive, HER2-positive breast cancer cases. Our analysis revealed that low ER expression (1-9%) and ER-negativity emerged as independent factors associated with pCR compared to high ER expression, as detailed in the revised Table 2 and 3. Notably, as anticipated, we observed a remarkable pCR rate in the low ER expression group within ER-positive, HER2-positive breast cancer cases, akin to the pCR rate observed in ER-negative, HER2-positive breast cancer cases.
In all patients and ER-positive, HER2-positive breast cancer, low ER expression (1-9%) and ER-negativity were independent factors associated with pCR compared to high ER expression (revised Table 2 and 3). Interestingly, as you expected, we observed outstanding pCR rate in low ER expression group in ER-positive, HER2-positive breast cancer similar to the pCR rate of ER-negative, HER2-positive breast cancer.
We have modified that all relevant sections throughout the manuscript reflect these revisions appropriately.
2) In your methods section you mention assessing Ki-67 labelling index on the patients' cancers, but you never mention it again. Is this because it did not impact pCR rates or the impact of HR status, HER2 expression, or TIL level on pCR rates? If so, this would be worth mentioning.
Answer) Thank you for your comment. Unfortunately, Ki-67 was investigated in only 60 patients (17.2%) of the whole cohort because this value was not routinely evaluated in core needle biopsy samples at our institution. In addition, as you mentioned, Ki-67 was not different according to ER status, HER2 expression, TIL levels, and pCR. However, as pointed out by Reviewer 1, we presented median and IQR valued for Ki-67 (revised Table 1). Also, we added the univariable logistic value of Ki-67 for pCR in all patients and ER+HER2+ breast cancer patients (revised Table 2 and 3). But, Ki-67 was not included in the multivariable logistic model because it was evaluated in a small number of patients. We presented related contents in the “limitation” of the “Discussion” as follows:
Third, Ki-67 is assessed in only 60 patients (17.2%) of the whole cohort because this value was not routinely evaluated in core needle biopsy samples at our institution. Accordingly, we excluded Ki-67 from multivariable analysis to maintain statistical power although Ki-67 is a well-established predictive factor for pCR.
3) There is a major typo on page 6. line 190 - you're reporting the impact of high or low TILs on pCR rates in HR-negative patients but you mistakenly repeat IHC3+ vs. IHC2+ instead.
Answer) We are sorry for the typos. We revised the manuscript according to your point.
4) Please increase the font on the legend for Figure 4 - this is your most important data - don't want readers to have trouble seeing it.
Answer) Thank you for your point. According to your point, we increased the font on the legend for all figures. Moreover, we modified the overall figures (revised figure 2-5), referring to other reviewers’ suggestions. In particular, the original Figure 4 was replaced with revised Figure 5, which shows the pCR rate according to HER2 protein expression and ER expression in ER+HER2+ breast cancer stratified by TILs.
Figure 2. Pathologic complete response according to (A) ER status, (B) HER2 immunohistochemistry (IHC) expression and (C) tumor-infiltrating lymphocytes (TILs). ER, estrogen receptor; HER2, human epidermal growth factor receptor 2
Figure 3. Pathologic complete response according to (A) HER2 immunohistochemistry (IHC) expression and (B) tumor-infiltrating lymphocytes (TILs) in ER-negative, HER2-positive breast cancer. HER2, human epidermal growth factor receptor 2; ER, estrogen-receptor. *p-value was obtained with the Fisher’s exact test.
Figure 4. Pathologic complete response according to (A) HER2 immunohistochemistry (IHC) expression, (B) tumor-infiltrating lymphocytes (TILs) and (C) ER expression in ER-positive, HER2-positive breast cancer. HER2, human epidermal growth factor receptor 2; ER, estrogen-receptor.
Figure 5. Pathologic complete response according to HER2 immunohistochemistry (IHC) expression and ER expression in ER-positive, HER2-positive breast cancer stratified by tumor-infiltrating lymphocytes (TILs). HER2, human epidermal growth factor receptor 2; ER, estrogen receptor. *p-value was obtained with the Fisher’s exact test.

Reviewer 3 Report
Comments and Suggestions for Authors
In this manuscript, the authors identified three biomarkers that may predict treatment outcomes for a neoadjuvant systemic therapy with docetaxel, carboplatin, trastuzumab, and per-17 tuzumab (TCHP). The first one is hormone receptor (HR)-status. If HR is negative, HER2-positive patients would very likely have excellent response to TCHP. On the other hand, if HR is positive, high HER2 protein expression levels indicated by immunohistochemistry (IHC) and/or high tumor-infiltrating lymphocyte (TIL) levels usually predict positive response for TCHP. In contrast, HER2-positive patients with low TIL and HER2 levels responded poorly to the TCHP treatment and it is necessary to find alternative therapies for them. The findings reported here have good clinical implications. The manuscript is well written and is relevant to the readership of the journal.
Minor points:
1. In line 20, would it be HER2-positive instead of HER-positive?
2. In line 101, it stated that “… carboplatin was administered at an area under the curve of 5 or 6 in all cycles.”. Could it be elaborated a little further regarding “an area under the curve of 5 or 6”?
3. In Tables 2 and 3, Ref. needs to be defined. Would it mean a control group?
Author Response
<Reviewer 3>
In this manuscript, the authors identified three biomarkers that may predict treatment outcomes for a neoadjuvant systemic therapy with docetaxel, carboplatin, trastuzumab, and per-17 tuzumab (TCHP). The first one is hormone receptor (HR)-status. If HR is negative, HER2-positive patients would very likely have excellent response to TCHP. On the other hand, if HR is positive, high HER2 protein expression levels indicated by immunohistochemistry (IHC) and/or high tumor-infiltrating lymphocyte (TIL) levels usually predict positive response for TCHP. In contrast, HER2-positive patients with low TIL and HER2 levels responded poorly to the TCHP treatment and it is necessary to find alternative therapies for them. The findings reported here have good clinical implications. The manuscript is well written and is relevant to the readership of the journal.
We are sincerely grateful for the time and thoughtful consideration you have dedicated to reviewing this manuscript. We acknowledge the constructive feedback provided by the reviewers, and we have incorporated many of their suggestions to improve clarity. Your valuable input has significantly contributed to enhancing the quality and precision of our manuscript. Thank you once again for your invaluable contributions.
Minor points:
- In line 20, would it be HER2-positive instead of HER-positive?
Answer) We revised the manuscript according to your point.
- In line 101, it stated that “… carboplatin was administered at an area under the curve of 5 or 6 in all cycles.”. Could it be elaborated a little further regarding “an area under the curve of 5 or 6”?
Answer) Thank you for your comment. This was about the dosage of carboplatin. Based on your suggestion, we have supplemented the information as follows:
Docetaxel was administered at 75 mg/m2 and carboplatin was administered at an area under the curve (AUC) of 5 or 6 in all cycles. The dosage of carboplatin was determined using the Calvert formula. The maximum dosage was calculated by an estimated glomerular filtration rate, which was capped at 125 ml/min for patients with normal renal function. The maximum doses were calculated using the following formula: Maximum carboplatin dose (mg) = target AUC (mg/mL/min) × (125 ml/min + 25). For a target AUC of 6, the maximum dose was 6 × 150 = 900 mg. For a target AUC of 5, the maximum dose was 5 × 150 = 750 mg. The physician determined the carboplatin dose based on the patient's general characteristics, including age and underlying diseases.
- In Tables 2 and 3, Ref. needs to be defined. Would it mean a control group?
Answer) You're correct. “Ref” denotes control group. To enhance clarity, "Ref" has been modified to "1 (Reference)".

Round 2
Reviewer 1 Report
Comments and Suggestions for Authors
The corrections of the manuscript are well done. In Figures 2-5 there is no y-axis legend. It should be added or at least be clearly explained in the text under the figure (percentage of patients with pCR).
Author Response
Answer) Thank you for your valuable comment. Based on your suggestion, we added y-axis legends in Figures 2-5.
